# Understanding successful development of complex health and healthcare interventions and its drivers from the perspective of developers and wider stakeholders: an international qualitative interview study

Katrina M Turner,[1] Nikki Rousseau,[2] Liz Croot,[3] Edward Duncan,[4] Lucy Yardley,[5,6] Alicia O'Cathain,[7] Pat Hoddinott[8]

For numbered affiliations see end of article.

**Correspondence to**
Dr Katrina M Turner;
Katrina.Turner@bristol.ac.uk

## ABSTRACT

**Objective** Identify how individuals involved in developing complex health and healthcare interventions (developers), and wider stakeholders in the endeavour, such as funders, define successful intervention development and what factors influence how interventions are developed.

**Design** In-depth interviews with developers and wider stakeholders to explore their views and experiences of developing complex health and healthcare interventions.

**Setting** Interviews conducted with individuals in the UK, Europe and North America.

**Participants** Twenty-one individuals were interviewed: 15 developers and 6 wider stakeholders. Seventeen participants were UK based.

**Results** Most participants defined successful intervention development as a process that resulted in effective interventions that were relevant, acceptable and could be implemented in real-world contexts. Accounts also indicated that participants aimed to develop interventions that end users wanted, and to undertake a development process that was methodologically rigorous and provided research evidence for journal publications and future grant applications. Participants' ambitions to develop interventions that had real-world impact drove them to consider the intervention's feasibility and long-term sustainability early in the development process. However, this process was also driven by other factors: the realities of resource-limited health contexts; prespecified research funder priorities; a reluctance to deviate from grant application protocols to incorporate evidence and knowledge acquired during the development process; limited funding to develop interventions and the need for future randomised controlled trials (RCTs) to prove effectiveness. Participants expressed concern that these drivers discouraged long-term thinking and the development of innovative interventions, and prioritised evaluation over development and future implementation.

**Conclusions** Tensions exist between developers' goal of developing interventions that improve health in the real world, current funding structures, the limited resources within healthcare contexts, and the dominance of the

### Strengths and limitations of this study

► Interviews were held with developers and wider stakeholders who varied in terms of their disciplinary backgrounds, seniority and role in developing interventions.
► The open, flexible nature of the interviews enabled participants to raise issues that were salient to them.
► The interviews were conducted by an experienced researcher who familiarised herself with each participant's work prior to interview, allowing her to ask relevant questions.
► While aiming to secure an international sample of participants, this proved difficult and only four interviews were conducted with individuals not living in the UK.
► To maintain confidentiality and anonymity, prior to data analysis interviews were anonymised and details of interventions, research studies, contexts and individuals removed. This may have reduced the richness of the data.

RCT for evaluation of these interventions. There is a need to review funding processes and expectations of gold standard evaluation.

## BACKGROUND

In 2008, the UK Medical Research Council (MRC) published guidance on how to develop and evaluate complex interventions.[1] Since then researchers have detailed different approaches to intervention development,[2–4] and a recent review of this literature identified 8 categories of approaches and 18 actions undertaken within them.[5] Other reviews have detailed how researchers have developed interventions in practice,[6 7] yet concerns have been expressed that too many

interventions are not found to be effective or are not implemented in the real world.[8]

This lack of effectiveness and implementation might be because despite increasing amounts of guidance on how to develop interventions, research to date has not discovered which approaches or specific actions are essential to develop interventions that will improve health outcomes. For example, systematic reviews of published approaches to intervention development have concluded there is a lack of evidence to determine their value in terms of future effectiveness. This is the case for co-produced or co-designed interventions,[9 10] and for intervention mapping.[11] There is some evidence that the use of theory within intervention development leads to effective interventions,[12] but there are also concerns that the relationship between theory use and effectiveness is weak,[13] and a recent review of reviews concluded that theory-based health interventions were not more effective than non-theory-based interventions.[14]

With uncertainty around which approaches to, or actions in, the intervention development process lead to success, researchers involved in developing complex interventions decide for themselves what the focus and content of their development process should be. Understanding how they define successful intervention development and what drives the decisions they make, could indicate their priorities and intentions, and subsequently provide new insights into why some interventions fail to have impact in the real world. It would also indicate how they define success and therefore what outcomes they think different actions in the development process should address. To date, the research that has considered the value of different approaches to, or actions within, intervention development has focused on the effectiveness of the interventions produced, but researchers may have other definitions of success they are working towards. They may, for example, judge the development process according to how it is undertaken, as well as against the impact of the resulting intervention.

The MRC-National Institute for Health Research (NIHR) funded a study to produce guidance on intervention development: IdentifyiNg and assessing different approaches to DEveloping compleX interventions (INDEX).[15] As part of this study, in-depth interviews were held with individuals developing interventions and wider stakeholders. This paper aims to detail how those interviewed defined successful development and the factors they viewed as driving the development process.

## METHODS
### Overall design
The design was a qualitative interview study of individuals who had developed complex interventions to improve health (ie, developers) and wider stakeholders. Wider stakeholders were individuals responsible for or affected by health-related and healthcare-related decisions that may be informed by research evidence, for example,

funding panel members, and patient and public involvement (PPI) members. The study was pragmatic in nature and used the fact that employing a qualitative method would enable attention to be given to context and processes, and for participants to describe their views and experiences in detail.[16]

The overall purpose of the interviews was to gain detailed insight into the challenges of developing complex health and healthcare interventions. The INDEX study they were part of, consisted of a systematic methods overview of different approaches to intervention development,[5] a review of journal articles reporting intervention development for specific studies, the qualitative interview study reported here, and a Delphi consensus exercise that the reviews and interviews fed into.

Throughout the INDEX study, the focus was on the development of complex interventions that could be delivered in public health, primary, secondary or social care settings. These interventions could include behaviour change interventions and interventions directed at health providers, patients and policy makers. It did not focus on medicines or invasive interventions, for example, pills or devices.

### Patient and public involvement
PPI members were invited to an expert panel meeting held at the start of the INDEX study, and to a consensus conference held once the Delphi results were known. None attended the first meeting, but PPI members did attend the consensus conference aimed at interpreting the Delphi results. All study participants will be sent copies of journal publications resulting from the INDEX study.

### Recruitment and sampling
For the qualitative interview study, we identified potential participants through various routes: the results of the two reviews conducted as part of the INDEX study, members of the INDEX project international expert panel, our personal knowledge of individuals working in health research and on funding panels, looking at funding websites and asking individuals interviewed to suggest others we might want to approach. Having identified potential participants, we purposefully sampled[17] individuals to obtain maximum diversity within our sample according to role within the development team (lead, co-applicant, stakeholder), professional background (clinical including medical, nursing and allied health professionals, social scientists, others), geographical location (UK, other European countries, North America), intervention type (eg, behaviour change, e-health, complex care packages) and setting (primary care, secondary care, public health and social care). We also aimed for diversity in terms of level of experience, sampling individuals who had completed just one intervention development, through to those who had contributed to multiple developments.

## Data collection

Potential participants were approached by email with an attached information sheet. The information sheet provided further details about the qualitative study, what participation would entail and included a copy of the consent form that would be completed prior to interview. The sheet also informed individuals that if they took part, they would not be named or identifiable in any way. Providing anonymity was viewed as important because some participants would be well known within the research community. Interested participants were contacted by NR to arrange a suitable time and mode of interview.

All interviews were conducted by NR, who has over 20 years' experience of conducting qualitative interview studies and evaluating healthcare technologies, but has limited experience of intervention development. This meant that she had a good understanding of the general context but probed on aspects of intervention development that might be taken for granted by a more experienced developer.

Before each interview, NR familiarised herself with the participant's published intervention development work, so that she could ask relevant questions, and participants completed the consent form (verbally for telephone/Skype and written for face-to-face). After each interview, NR made reflective field notes to assist with analysis.

A topic guide was used to ensure key areas were covered, while the open, flexible nature of the interview process allowed participants to raise issues that were personally salient. The guide was informed by the overall aim of the qualitative study and early findings from the INDEX systematic review of approaches to intervention development.[5] Key areas covered by the guide included participants' use of intervention development approaches, their experience of developing interventions, how they had developed and evaluated specific interventions, and what guidance and advice they would give to researchers undertaking this process. Initially, the guide did not include specific questions about how successful intervention development should be defined, and what factors shaped and drove the development process, but analysis of the first few interviews indicated definitions of success and drivers were important themes in the data. Thus, after the first few interviews, the following questions were inserted into the guide: "I'm interested in what counts as success in the context of intervention development. Do you think of this development study, as a successful study? Why?; Were there any key aims or principles guiding the development of the intervention? What and why?"

Early on during data collection, the importance of maintaining participant anonymity was confirmed; one person reviewed their transcript and asked for a small amount of text to be deleted, another person asked during the interview for specific comments to be 'off the record'. Following these interviews, the team took the decision that only NR would know who had been interviewed.

## Data analysis

All the interviews were audio-recorded, then transcribed verbatim by an external transcribing service, and checked and anonymised by NR. NR removed details of individuals, places and interventions before sharing transcripts with other team members.

The data were analysed thematically,[18] so that comparisons could be made within and across the interviews, and participants' views of specific issues highlighted, for example, how individuals identified the need for a new intervention. To do this, a coding frame was drafted, tested and refined. NR (social scientist), PH (Academic General Practitioner), KMT (social scientist) and ED (health service researcher and allied health professional) independently read three transcripts and constructed a coding frame. They then discussed their coding and combined their coding frames into a single framework. Each of them then applied this framework to two new transcripts from interviews conducted at different times in the data collection process. Another discussion was then held which resulted in the addition of further codes, and the deletion or clarification of some existing ones. The framework was then finalised, and applied by NR to transcripts using NVivo 11 software.[19] Each member of the team applied the final coding frame independently to one transcript; the aim of this exercise was not to establish inter-rater reliability as all subsequent coding using this framework was conducted by NR, but to allow comparison of, and reflection on, differences.

The framework applied by NR included the code 'definitions of success' and other codes that were relevant to the aims of this paper, for example, development pathway, study end point, evaluating complex interventions. NR, KMT and PH read data under these codes and discussed themes they had identified. As the framework NR applied had not been developed with the specific aims of this paper in mind, following these discussions KMT re-read all the transcripts to check whether there were other relevant themes which needed to be captured. This also helped her contextualise and fully understand comments made. The re-reading of these transcripts led to KMT developing a few more codes, for example, indicators of success, actions needed, tensions. These codes were added to the framework used by NR and applied by KMT where appropriate. Data coded under them were then discussed with NR and PH. Using an approach based on framework analysis,[20] data relevant to this paper were summarised in a table formatted so that the rows represented each participant and columns relevant codes. KMT then read and re-read the table's content, reading down each column in order to identify similarities and differences between the views and experiences of different participants in relation to a specific code or theme, and across the columns to consider what might have influenced each participant's views and experiences.

Throughout the analysis phase, the emerging analysis was discussed at regular team meetings (NR, PH, KMT and ED) and with AO'C. At these meetings, NR and KMT

**Table 1** Participants' details (n=21)

| | Number |
|---|---|
| *Disciplinary background* | |
| *Clinician (doctors, nurses, allied health, public health)* | 10 |
| Other (health psychologist, health economists, sociologists, product design) | 9 |
| *Patient* | 2 |
| *Country* | |
| UK | 17 |
| Other parts of Europe | 2 |
| North America | 2 |
| *Gender* | |
| *Female* | 13 |
| Male | 8 |

detailed their insights and interpretation of the data, inviting other team members to comment in light of the aims of the analysis and their knowledge of the data and literature on intervention development. Data collection ceased when no new themes were being identified or significantly elaborated, that is, when data saturation[21] had been reached.

## FINDINGS
### The participants

The interviews were conducted between February 2017 and January 2018. Twenty-nine individuals were invited for interview and 21 agreed (table 1). Most of those who did not respond to the invitation were outside the UK.

Six of the 21 participants interviewed were wider stakeholders. Three of these stakeholders were funding panel members, two were PPI members and the remaining stakeholder worked in health service implementation. In terms of the remaining participants, that is, the developers, eight had led intervention development, four as 'senior leads' and four as early career researchers. Some of the senior leads had contributed to over 20 development projects. However, experience was about number of interventions completed and about the size and complexity of these interventions. Senior leads had typically been responsible for developing highly complex interventions that would need considerable health service buy-in to implement, and had developed these interventions in the context of large scale, £1 million plus (or equivalent in other currencies) research funding. Early career researchers had typically completed smaller scale developments as part of a doctoral or postdoctoral fellowship. Participants who had developed the most interventions typically had specialist skills, for example, product design or behavioural science.

Except for one participant who had developed an intervention mainly on their own, participants had worked within a team, and most participants had held various roles across different intervention developments (eg, lead on one, co-investigator on another). Only two participants (both on research funding panels) had no direct experience of developing interventions. Between them, participants had experience of a wide range of intervention development approaches, including theory based, partnership, target population-centred and combined approaches.[5]

Seventeen of the interviews were conducted by telephone, two by Skype (with video) and two (both PPI contributors) on a face-to-face basis. They lasted, on average, 1 hour.

### Overview of findings

Findings are detailed below under two main headings: 'successful intervention development' and 'drivers of intervention development'. This structure reflects the aims of the paper, but these sections relate and overlap, as some drivers reflected participants' definitions of success, and developers' views of success could drive the development process.

### Successful intervention development

Participants' definitions of successful intervention development related both to the characteristics of the resulting intervention, and to how they thought the actual development process should be undertaken. The definitions they gave suggested they thought about short-term, medium-term and long-term goals, all of which they might consider from the start of the development process but for which evidence to indicate they had been achieved would come at different time points in the future (figure 1).

#### Characteristics of the resulting intervention

Most participants talked about aiming to develop effective interventions. Effective interventions were defined as those that reduced the prevalence, incidence or the implications of a health condition; improved public and patient understanding, treatment outcomes and the quality and delivery of care and care systems; and reduced healthcare costs by being more cost-effective than current care, ensuring more appropriate referrals or reducing unnecessary treatments. Yet participants also talked about wanting to develop interventions that were relevant to current care systems, health contexts and policies; acceptable to practitioners, populations, patients and individuals who commissioned healthcare, and feasible to deliver in resource-limited health and social care systems (UK participants only). Developing an intervention that was not practically or financially feasible was described as wasting public money:

*If somebody says to me, I want to train two educators, I want to deliver to 20 patients over a year, I go, 'well that's very nice, so when you've got a real plan come back, because this is public money'. Very conscious of that. We're very conscious about being in there for the long term because there is an*

| Definition/measure | Short term definitions relating to processes and outcomes | Medium term definitions relating to outcomes | Long term definitions relating to outcomes |
|---|---|---|---|
| **Intervention** | Acceptable to stakeholders<br><br>Feasible to deliver<br><br>Meets the needs of end users<br><br>Relevant to health contexts, practices and polices | Effective: improves health, public and patient understanding, changes practice<br><br>Cost-effective<br><br>Changes thinking | Implemented in the real world<br><br>Transferrable to different contexts<br><br>Sustainable in the real world<br><br>Scalable to reach all relevant populations |
| **Development process** | Uses research evidence<br><br>Incorporates stakeholders' views<br><br>Demonstrates methodological expertise<br><br>Scientifically robust | | |
| **Academic impact** | Journal publications<br><br>Professional credibility, particularly with funders | Future grant funding | |

**Figure 1** Definitions of success.

*investment.* (Interviewee 11, wider stakeholder, health service implementation)

While a few participants mentioned that some researchers wanted to demonstrate the effectiveness of their interventions before thinking about how they would be implemented on a large scale, most commented that it was important to think about long-term sustainability right from the start of the development process. In addition, several participants mentioned that transferability and scalability of the intervention to different care systems and cultures also needed to be considered early on. Yet, despite this focus on ensuring potential widespread use, one participant made the point that an intervention could be viewed as worthwhile if it changed thinking in relation to a health issue, even if it was not widely implemented or kept in its original form.

*I don't think it (the intervention) was adopted widely … I don't know why… perhaps it was because hospitals are so busy crisis managing. I think the success has probably been is that a lot of other hospitals looked at (the intervention) and thought we can adapt this… it's been adapted and rippled out… So it's acted as a game changer in the fact that it's changed the way other people will look at (the health condition)… that's how I think (the intervention) would be the most effective.* (Interviewee 1, wider stakeholder, PPI)

As well as judging whether the development process had been successful according to the intervention's future outcomes and impact, a few participants made comments that indicated successful intervention development could also be defined as one where the resulting intervention met the needs and expectations of others. For example, one participant who was particularly interested in real world implementation of proven interventions, defined successful development as one that resulted in an intervention which reflected what patients and practitioners needed and wanted. This definition was clearly linked to his other definition of success, which was to develop interventions that were *'useful, appropriate, acceptable, applicable and scalable within practice settings'* (interviewee 4, wider stakeholder, funding panel member), in that he believed if it did not meet the needs of these individuals, it would not be implemented or used. Other participants also made this point.

### A well-conducted research process

Some participants commented that the development process should be a robust piece of research that was methodologically strong and incorporated existing evidence and stakeholders' views. One of these participants argued that it was important to demonstrate a robust process alongside developing interventions that were clinically relevant, and that these different measures of success were not conflicting.

*I think the other thing is ensuring that the idea, the research, has clinical credence, that it's going to be useful… And I don't think the different ways of measuring success, the clinical way of measuring success - patients are better, patients are happy, they accept intervention. The academic measures of success with academic colleagues are that it has to be robust, and it shows we know what we're doing and we get support from the (funding) board. I think if people sit down*

*and understand that those two different aims aren't conflicting, which I don't think they are—they shouldn't be—then it can work nicely.* (Interviewee 8, developer)

Lastly, a couple of participants, one who was a very experienced developer and the other less experienced, commented that a successful development process would also produce research evidence on which to base journal publications and secure future research funding.

### Drivers of intervention development

Some participants described how their development processes had been driven by a single factor, while others had been shaped by multiple drivers operating at different levels or at different times during the development process. These drivers could be organised under five main headings: developers' ambitions; existing health contexts, care and legislation; research evidence, theories and methods; securing funding and the need to evaluate using randomised controlled trials (RCTs).

### Developers' ambitions to improve health outcomes

When detailing where the initial idea for an intervention had come from, developers described how they had identified through their work a specific health or healthcare issue they wanted to improve. It was clear this ambition to improve a situation had been their motivation to develop the intervention, and that the issue they had identified had driven the intervention's focus.

*What I'm really interested in, it's about how we get really, really good evidence-based medicine and access to those services for socially marginalised groups… which de facto means complex interventions…* (Interviewee 14, developer)

This aim of wanting to make a difference fits with the aim of developing effective interventions for real-world contexts, which in turn requires attention to be given to the context in which interventions will be used, making health contexts another driver.

### Existing health contexts, care and legislation

Participants explained that they needed to develop interventions that were feasible and affordable to deliver within populations, and within existing health contexts, practices and systems. Some participants also talked about needing to secure support from individuals who would be delivering the intervention, or from managers or purchasers of care who could prevent it from being provided. As participants working in the UK talked about practitioners not having the time and resources to do more, it did seem that the need to consider the realities of health contexts and secure support from others, meant developers had to limit how resource-intense their interventions could be.

*… what you see often in both medicine and therapies, they have a fantastic approach delivered, but it takes an hour more than your clinic slot allows, and therefore it won't be adopted because it's just not practical or financially feasible… to be clinically credible is important, but also I knew that management and the commissioners (of care) would never countenance it, would never allow it to happen, if it was going to cost and take three times as long as normal.* (Interviewee 8, developer)

It was apparent that existing care and legislations could also drive what the intervention would be, again by indicating what would be considered possible and acceptable.

*So (the intervention) is recommendation-based models of care that are implemented in all (European country) hospitals or almost all (European country) hospitals. And it's part of the legislation that was introduced in 2007.* (Interviewee 15, developer)

Yet alongside this need to consider health contexts, accounts suggested that during the development process, attention also needed to be given to academic or research contexts. It seemed that developers looked to health contexts to ensure their interventions were relevant and acceptable, but to research evidence, theory and methods to ensure their effectiveness.

### Research evidence, theories and methods

Most participants described how the content and structure of their interventions had been informed through published evidence, which had indicated areas and issues requiring careful consideration to ensure future effectiveness. A few participants also talked about using theories to gain a deeper understanding of an issue and to guide their choice of intervention components. Choice of components could also be based on the participant's previous research, and again selected with the aim of enhancing the intervention's potential effectiveness.

*The way our intervention was developed was we had some initial ideas based on some of our, I'll say, effectiveness work on single components or a few components before… we put a number of things together.* (Interviewee 18, developer)

The use of existing evidence, theories and historical experience meant that key components of an intervention were agreed from the start of the development process. Alternatively, reviewing existing evidence could be part of the development work, and could result in the intervention being developed in ways that had not been initially predicted.

*There definitely was a shift, definitely was a shift in the sense that the evidence statements, from the reviews that identified the behaviour change techniques that were more likely to be associated with effectiveness, we wouldn't have known that in the bid, so that was a complete unknown, we could've supposed what might be the likely candidates from the other literature but that was definitely new.* (Interviewee 2, developer)

Similarly, participants talked about how discussions with stakeholders, PPI members, topic experts and the results of primary research conducted as part of the

development study, had also driven the development process in new directions. However, in some circumstances participants described how they had ignored such insights because they believed funders would want them to develop the intervention they had prespecified, there were not the resources to develop the intervention in the new ways suggested, or because the Principal Investigator on the study had very fixed ideas about what the intervention should be.

> Interviewee: *The intervention was always going to be a nurse. It didn't matter what we'd found out about what people wanted or what was missing, the answer was going to be a nurse.*
>
> Researcher*: So it sounds like this project's been designed so that you will do some qualitative research to develop the intervention but alongside that there'd already been a prior unacknowledged…*
>
> Interviewee: *Well I suppose it's acknowledged cause it was in the bid…That I think is the other big problem you have to specify your intervention in the bid, you know what it is.* (Interviewee 21, developer)

Very few participants talked about using a formal approach to intervention development. Some individuals did mention using the MRC framework,[1] but this was described as not providing enough detail to guide the development process or paying enough attention to implementation.

### Securing funding to develop interventions

Several participants, including those who were on funding panels, commented there was more funding available to evaluate interventions than to develop them, and little or no funding to facilitate implementation of interventions.

The way developers applied for funding created further perceived barriers around successful development. Participants explained that researchers usually needed to apply to different funders, for individual pots of money, to fund different stages of the development and evaluation process. Although one participant argued this approach gave flexibility which probably led to a better developed intervention, this participant and others explained that this situation extended the time between the initial idea and real-world implementation, a period in which contexts may change, rendering the intervention ineffective or irrelevant. It also meant team members changed, losing valuable knowledge and experience, and could stop the development process if funding for the next phase was not secured. UK participants described NIHR Programme Grants for Applied Research as allowing both development and evaluation, but even 5 years of funding was viewed as insufficient time to do both.

> I think, the model where you get in the one package the funding where you do your pilot, your feasibility, intervention development, and your piloting, and then your trial, and you know that, as long as it's all going well, they're going to keep funding you, is probably the only way to go. But, even

*5 years on a programme grant isn't long enough to do that.* (Interviewee 13, developer)

Where researchers had secured funding in response to commissioned calls, that is, calls where the funder had specified the research question to be addressed, or known the priority areas of a funder, this had determined the focus of their intervention. It was also apparent that funders could dictate what the intervention should be, and funding briefs determine what the development process entailed and what needed to be delivered.

> *(the intervention) was a thing the (government) wanted… and we implemented it.* (Interviewee 5, developer)
>
> *We knew we had to start some iterative pilot testing… and we knew that we had to try to get to a phase of something that was testable because the mandate from our funder.* (Interviewee 18, developer)

When describing the process of writing grant applications, developers talked about playing it safe and aiming to develop interventions that the funders could relate to, rather than interventions that were innovative and required new ways of thinking. In addition, although funding board members interviewed said there was flexibility in terms of the extent to which the potential intervention needed to be defined in the grant application, one developer described how she felt she had needed to specify the intervention and the development process, even though this was unrealistic.

> Researcher: *So are you saying that it would have been okay to have said, 'we think some sort of technology in the home?'*
>
> Interviewee: *Would have never got funded. It had to be reified, specific, and we had to be able to say, 'it's this gadget, this soft(ware)…' I had to be all rational… normative… All that stuff that you pretend is true when you put in a proposal… it's a figment of everybody's imagination, it's created… The only places that work like that are nuclear power stations, aeroplanes, and a few other things, because you've got a controlled environment. Yeah. Unless you have a controlled environment, ideas of linear, normative planning are ridiculous.* (Interviewee 5, developer)

### The need to evaluate using RCTs

Participants talked about public health and healthcare practitioners, and purchasers of healthcare services, wanting RCT-based evidence that demonstrated the intervention's effectiveness and cost-effectiveness before they were willing to support or purchase it. This meant developers felt a need to develop interventions that could be evaluated within a RCT. Several participants viewed this as problematic, explaining that they felt there was a tension between developing interventions that were suitable for use in the real world and developing interventions that could be evaluated in an RCT. It was argued, for example, that the more testable an intervention, the less real it became, and that protocolising an intervention for research purposes reduced its practicality. Thus, this drive

for evidence could work against the goal of developing useful real-world interventions.

> *The work you have to do to turn the intervention into something that's testable. Which of course, the more you turn it into something that can be subject to an evaluation, the less real it becomes, so its characteristics become more specific rather than generic.* (Interviewee 5, developer)

> *It needed to be flexible enough to meet the patient's needs. But, having a very strict protocol… although it makes it more scientifically credible, isn't practically deliverable in a clinic setting.* (Interviewee 8, developer)

Two participants described the MRC framework as encouraging the push to develop interventions that could be evaluated within RCTs. They described it as '*detailing the different phases you go through to create a trial of a complex intervention… it is all about how do you create a complex intervention trial*' (Interviewee 5, developer) and as being focused on '*you should do that RCT*' (Interviewee 15, developer).

## DISCUSSION
### Principal findings
Developers and wider stakeholders defined successful intervention development as a rigorous, scientific process that resulted in effective interventions that could be implemented in real-world contexts. The drivers that shaped how interventions were developed indicated tensions arose when trying to meet the needs of existing structures. Developers targeted areas where change could result in better health outcomes. Research evidence and theory informed the content, structure and delivery of interventions to increase the likelihood of them being effective. The characteristics of health contexts also shaped the structure and content of the interventions developed, although here the drive was mainly to ensure they would be acceptable and feasible to deliver. Yet, as UK participants talked about how resource limited these health contexts could be, this driver also encouraged developers to be pragmatic and realistic in their designs, developing interventions that were not innovative or resource-intense, possibly reducing their potential effectiveness. Perceived requirements of funding bodies also discouraged innovation, and the reliance on RCTs to prove effectiveness worked against the need for real-world relevance.

### Strengths and weaknesses of the study
The in-depth and flexible nature of the interviews with a diverse sample of intervention developers and stakeholders allowed participants to describe, in detail, issues important to them. However, we struggled to recruit individuals living outside the UK which could limit the generalisability of our findings, particularly in relation to funding and healthcare context, although there did not appear to be any clear differences between the views of our UK and international participants about the

development process. The need to maintain anonymity promised to each participant did mean details of individuals, places and interventions needed to be removed prior to data analysis. This might have reduced the richness of the data but, based on reactions of some participants, anonymity encouraged individuals to talk openly about their experiences.

### Strengths and weaknesses in relation to other studies
To our knowledge, this is the first study to explore developers' views of successful intervention development, although definitions of success have been implied within the literature by researchers aiming to improve the development process. For example, Bleijenberg *et al*[22] have proposed a development approach that 'gives developers a better chance of producing an intervention that is well-adopted, fits its context, is effective and ready for piloting and trialling', (p. 92); a statement which echoes the main definition of success detailed here.

The importance of integrating health and research contexts early on, and considering future implementation and sustainability, are acknowledged as important in the development literature.[23–25] However, little is known about how many new interventions move on to formal piloting, definitive testing and then implementation into practice.[26] The current evidence suggests the number is low,[8] overconfidence prevails and a range of biases operate.[26] Many developers still view RCTs as the only way to evaluate an intervention, but this view is being challenged as researchers acknowledge other methods could be used to assess effectiveness and causality.[27]

Hoddinott[26] comments that UK funding bodies have not prioritised intervention development. Inadequate development of an intervention reduces the likelihood of it having an affect,[28] and optimising the development of complex interventions prior to their evaluation is a way to reduce research waste.[22] The process of developing interventions might be improved if developers used guidance provided by formal intervention development approaches.[5] Our participants seldom mentioned such approaches, and this lack of engagement might reflect the lack of evidence about which ones lead to success.

### Implications for practice and future research
The findings of the INDEX study, along with the revised version of the MRC guidance on developing and evaluation complex interventions due to be published in 2019, should support developers in reflecting on and hopefully achieving successful intervention development. However, the process of determining what needs to be done will remain unclear until researchers provide evidence that demonstrates what actions lead to interventions that improve health. This should be a priority area for future research. In addition, funders need to consider giving more funding to intervention development research and how funding processes can better support innovation and long-term planning.

**Author affiliations**
[1]Bristol Medical School, University of Bristol, Bristol, UK
[2]Newcastle University Faculty of Medical Sciences, Newcastle upon Tyne, Newcastle upon Tyne, UK
[3]MCRU, School of Health and Related Research (ScHARR), University of Sheffield, Sheffield, UK
[4]NMAHP-RU, Stirling, UK
[5]Psychology, University of Southampton, Southampton, UK
[6]School of Health Sciences, University of Bristol, Bristol, UK
[7]School of Health and Related Research (ScHARR), The University of Sheffield, Sheffield, UK
[8]Nursing, Midwifery and Allied Health Professional Research Unit, University of Stirling, Stirling, UK

**Acknowledgements** The authors would like to thank all those who were interviewed as part of this study.

**Contributors** AO'C, PH, LC, ED, KMT and LY designed the study. NR conducted the interviews detailed in this article, which were then analysed by KMT, with support from PH and NR. KMT prepared the first draft of this paper. Later drafts were commented on by the other authors. All authors approved the final version of the article.

**Funding** This work was supported by the Medical Research Council grant number MR/N015339/1.

**Competing interests** None declared.

**Patient consent for publication** Not required.

**Ethics approval** Ethical consent to undertake the qualitative component of INDEX was secured from the University of Stirling, General University Ethics Panel (GUEP37).

**Provenance and peer review** Not commissioned; externally peer reviewed.

**Data sharing statement** Participants were not asked to agree to their data being available for other studies, as many were well known within the health research community and this would have deterred some from taking part in the study.

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
