## [Reviewer comments · BMJ Open]

ARTICLE DETAILS

TITLE (PROVISIONAL)	Understanding successful development of complex health and health care interventions and its drivers from the perspective of developers and wider stakeholders: an international qualitative interview study
AUTHORS	Turner, Katrina; Rousseau, Nikki; Croot, Liz; Duncan, Edward; Yardley, Lucy; O'Cathain, Alicia; Hoddinott, Pat

VERSION 1 - REVIEW

REVIEWER	Heather Colquhoun University of Toronto, Toronto, Canada
REVIEW RETURNED	10-Feb-2019

GENERAL COMMENTS	Thank you for asking me to review this paper. I enjoyed reading it and was pleased to see work being done in this important area. This study is interesting in that it aims to collect information from people who develop complex interventions in health and health care delivery contexts, about the processes of intervention development. I found the paper valuable and it contained a lot of useful and interesting information. I did find many areas in the paper unclear (see below) that made some aspects of the paper difficult to fully understand and appreciate. Please see my comments on areas that I thought needed elaboration and clarity. Most of them will extend beyond the specific section that I address them in (eg changes to the methods/aims could change information in the findings and/or the discussion). I do think the work is important and encourage consideration of my comments. The title and abstract need to be more indicative that you are looking at complex health and healthcare delivery interventions. An overall comment: It was unclear in many areas whether the paper was focused on the process of intervention design or the outcome of the intervention design processes (ie the intervention), or both. Or even also about intervention evaluation (the need for RCT's). Were the definitions of success about what makes a successful intervention or about what makes for successful intervention development? I can see the subtlety between the concepts (one leads to the other) but the paper often mixes these concepts. Clarity in the boundaries around what was being investigated here would help make the findings more useful. The second paragraph of the background is a good example of this
--

issue. The overall intent of the paragraph is unclear as it has information on both intervention development as well as intervention evaluation.

The paper seemed to have many very valuable perspectives but was often not 100% clear. Abstract results, sentence starting "Accounts...". I was not sure what this was saying. Are you saying that your participants wanted a rigorous design process that had the end result of something users wanted in the short term, and in the long term they wanted a publication and another grant? Is the short term success that one should involve users in development or that the resulting intervention should be something they want?

Using the term 'developers' (abstract and background) was not clear enough to know who you were describing and studying. Also, it would be helpful to establish earlier in the paper that these interventions are complex health and healthcare delivery interventions (or however you might word this).

Methods:

The methods should begin with a succinct review of the overall design. This might also be a good place to provide information on the larger study (I think this is INDEX) and your study, and how the 2 relate.

Sampling: Perhaps a better description of the main study (INDEX) would help clarify who were the interviewees. I was not clear what was meant by 'wider stakeholders in the endeavour'. And whether these were people who had developed and tested, and published on interventions? – ie how did you know they developed an intervention?

It would be helpful to clarify that these interventions to change health are in the context of health behaviour (smoking cessation) as well as interventions directed at health providers/patients/policy makers for the purpose of ensuring the care delivered was in line with best practices (following clinical practice guidelines). Also, what would define novice vs experienced? What is meant by intervention type and setting?

Methods: the interviews were conducted by an experienced health services researcher – this is good but what specifically is meant by this? What kind of experience did this person have?

You indicated the guide was based on the aims of the study and relevant literature – it would be useful to have more details here – what exactly was the guide based on? Presumably you mean this study as opposed to the INDEX main study?

I thought the types of questions asked were extremely useful but I had a few questions: A topic guide was developed to 'ensure key areas were covered', but what were the key areas? One of the aims of the study (end of background) was to determine how they defined success so it seemed strange that this was not initially included in the guide – why not and what was initially included in the guide? Some sample questions here would be useful. Perhaps even relating the guide to the 2 specific aims for the study – ie how they defined success and the factors that influenced the process they chose.

I was curious about the issue your interviewees had regarding anonymity. Was this determined after they agreed to participate or did this play a role in recruitment?

Analysis: Again, more detail is needed. How did you test and refine the initial coding scheme? It seems this was done prior to applying it to an interview.

What was meant by 'how individuals identified areas for intervention' - is this about the process of intervention design or about the health context that was being investigated, and if about the health context, how does this relate to the study aims of looking at the process of intervention design?

I am not sure how a thematic analysis allows you to compare across interviews. Typically a thematic analysis allows you to consolidate the main messages being found across all of the interviews.

I was not 100% certain what happened with the initial coding scheme once applied to the transcripts. Did you entertain changes to the scheme after the first few transcripts? I would like to see more about how the 'discussions at team meetings on the emerging analysis' shaped the final analysis.

Findings: I would like to see more detailed information on the participants. Who were the intervention developers vs the stakeholders? How were these each defined, and were there any differences in these in the results?. 'Most participants held various roles across different intervention developments' - this requires more detail. Did they develop an intervention alone, as part of a team, what specific experiences did they have. How many interventions? Did they test these interventions? Publish? How many of them follow specific development processes like intervention mapping or using the TDF and behaviour change wheel processes, MRC guidance? What is a 'target population centred' approach? Were any of them implementation scientists? There is not enough information on the participants.

Are the 'drivers of intervention development' the same as asking people about the factors that influence their processes (2nd study objective)? I think I would prefer the 'factors that influence how they develop their interventions'.

The first theme is really interesting but it seems like a mix between the outcomes of intervention development (eg. must be cost effective), and the process itself (eg. consider sustainability early). This theme seems more about what a successful intervention is as about, as opposed to successful intervention development. I think some conceptual clarity on this would help.

I did not follow how ambition to improve health was a factor in their processes. Wouldn't most people who develop these interventions do so in order to improve health or health care delivery? This seemed more like the way in which people ended up being interested in health services research, not the factors that influence how they design these interventions once they decide to do so. It seems more about why they do it, as opposed to how.

The section on existing health contexts, care and legislation seems nicely aligned with the aims of the study and quite interesting to read, the other themes focused on the process of development were also very informative and useful (and aligned with my assumptions about what the study was about). The issue of relying on RCT's is again, an important factor but it is not connected to development per se (unless you were able to outline how this imperative for evaluation has an effect on how they designed the intervention).

	Discussion: The principal findings should be revised given points made above for the findings, and could be shortened, giving more room to look at other literature. Weaknesses of the study: I was not sure what was meant by the interview process allowing interviewees to say what they thought, as opposed to the research team. Also it would be wise to indicate that a limitation is only have one coder (if I am right about that). Diagram 1: I was not fully able to interpret the diagram. How were short, medium and long term determined? Why would acceptability to stakeholders only be a short term goal? Is cost effectiveness only medium and not long? The reporting checklist indicates that grounded theory was used in the design, but this is not mentioned anywhere in the paper itself. Also, that 4 coders were used but this was only for one transcript, it seems that most of the coding was done by 1 person (but I could be wrong).
--	---

REVIEWER	Dr Joy Probyn University of Salford, UK
REVIEW RETURNED	22-Feb-2019

GENERAL COMMENTS	This is an interesting study that certainly makes a contribution to a gap in knowledge and will be a useful resource for researchers working in intervention development. There are some aspects of the paper which I feel merit improvement - all related to the methodology, outlined as follows: There is no reference to qualitative epistemology/theory as justification for the methods chosen. You say on the COREQ checklist that it was a grounded theory study - this needs to be explained and linked to the explanation of methods. There are no references at all in the methodology section. There needs to be a better explanation of how the interviews were designed and how this links to data analysis. It appears that the data analysis was deductive - i.e. the two themes were predetermined and the data was 'fitted' in to these categories. This does not fit with a grounded theory approach. The findings read as quite obvious/descriptive for this reason - and theme 2 seems to be more inductively generated than theme 1, with the use of sub-headings. The two themes are actually very similar and I think it would improve the work to conduct the analysis again, inductively, and present the themes in this way. This would add rigour to the study. For example, some inductive themes could be 'sustainability', 'dependability' etc. A deductive analysis is fine - but it needs to be explained in the methodology more clearly if so. There needs to be an explanation of how informed consent was obtained. Was there any involvement from the public in the design and conduct of the research? P7 44-45 suggests a grounded theory influence - explain.
---

	Where did the coding frame come from? P 38 - Reference needed for data saturation. P8 36-37 Why did people decline you say they were based outside of the UK but was this the reason they gave or your assumption? It shouldn't matter with the use of Skype? P8 54 What was the geographical spread of those invited?
--	--

VERSION 1 – AUTHOR RESPONSE

Reviewer: 1

Reviewer Name: Heather Colquhoun

Institution and Country: University of Toronto, Toronto, Canada

Please state any competing interests or state 'None declared': None declared.

Please leave your comments for the authors below

Thank you for asking me to review this paper. I enjoyed reading it and was pleased to see work being done in this important area. This study is interesting in that it aims to collect information from people who develop complex interventions in health and health care delivery contexts, about the processes of intervention development. I found the paper valuable and it contained a lot of useful and interesting information. I did find many areas in the paper unclear (see below) that made some aspects of the paper difficult to fully understand and appreciate. Please see my comments on areas that I thought needed elaboration and clarity. Most of them will extend beyond the specific section that I address them in (eg changes to the methods/aims could change information in the findings and/or the discussion). I do think the work is important and encourage consideration of my comments.

Thank you for these positive comments and for your helpful suggestions regarding how the manuscript could be improved.

The title and abstract need to be more indicative that you are looking at complex health and healthcare delivery interventions.

The title of the manuscript has been changed to 'Understanding successful development of complex health and health care interventions and its drivers from the perspective of developers and wider stakeholders: an international qualitative interview study'. It therefore now indicates we were looking at complex health and health care interventions.

In addition, we have changed the Abstract to make it clearer that we were focusing on complex health and health care interventions, i.e. under the subheadings Objective and Design we now state the focus was on the development of complex health and health care interventions.

An overall comment: It was unclear in many areas whether the paper was focused on the process of intervention design or the outcome of the intervention design processes (ie the intervention), or both.

Or even also about intervention evaluation (the need for RCT's). Were the definitions of success about what makes a successful intervention or about what makes for successful intervention development? I can see the subtlety between the concepts (one leads to the other) but the paper often mixes these concepts. Clarity in the boundaries around what was being investigated here would help make the findings more useful. The second paragraph of the background is a good example of this issue. The overall intent of the paragraph is unclear as it has information on both intervention development as well as intervention evaluation.

This is a very helpful comment to improve the clarity of our papers for readers. Participants' definitions of successful intervention development related to outcomes from the development process, i.e. the characteristics of the resulting intervention, and to how the development process was undertaken, e.g. the developmental process should incorporate existing evidence. The papers therefore details development goals relating to both the development process and to the resulting outcomes.

To indicate that we were open to definitions relating to both outcomes and processes, in the Introduction we have inserted the following text: 'To date, the research that has considered the value of different approaches to, or actions within, intervention development has focused on the effectiveness of the interventions produced, but researchers may have other definitions of success they are working towards. They may, for example, judge the development process according to how it is undertaken, as well as against the impact of the resulting intervention.'

To improve clarity within the Findings section, we have added the following text under the heading Successful intervention development: 'Participants' definitions of successful intervention development related both to the characteristics of the resulting intervention, and to how they thought the actual development process should be undertaken.' We have also inserted two subheadings within this section: 'Characteristics of the resulting intervention' and 'A well conducted research process'. In addition, we have moved a paragraph that summarised the data relating to participants' definitions of success, and which had been placed at the end of the section on Successful invention development, to the start of the Findings sections as a way of signposting readers to the various definitions given. We have also moved diagram that related to this summary. Lastly, we have made changes to the text given under the heading of Successful intervention development to make it clearer that these definitions of success were ambitions or goals participants had when undertaking the development process.

In terms of the second paragraph in the background, this paragraph has been edited to make it clearer that its purpose is to explain that there is a lack of evidence about which actions or steps within the development process are needed to ensure the resulting intervention will be effective and suitable for real-world implementation. It now reads: 'This lack of effectiveness and implementation might be because despite increasing amounts of guidance on how to develop interventions, research to date has not discovered which approaches or specific actions are essential to develop interventions that will improve health outcomes. For example, systematic reviews of published approaches to intervention development have concluded there is a lack of evidence to determine their value in terms of future effectiveness. This is the case for co-produced or co-designed interventions,^{9 10} and for Intervention Mapping.¹¹ There is some evidence that the use of theory within intervention development leads to effective interventions,¹² but there are also concerns that the relationship between theory use and effectiveness is weak,¹³ and a recent review of reviews concluded that theory-based health interventions were not more effective than non-theory-based interventions.¹⁴'

The paper seemed to have many very valuable perspectives but was often not 100% clear. Abstract results, sentence starting "Accounts....". I was not sure what this was saying. Are you saying that your participants wanted a rigorous design process that had the end result of something users wanted in the short term, and in the long term they wanted a publication and another grant? Is the short term success that one should involve users in development or that the resulting intervention should be something they want?

Thank you for this comment. We agree the sentence is not clear. It has been changed to 'Accounts also indicated that participants aimed to develop interventions that end users wanted, and to undertake a development process that was methodologically rigorous and provided research evidence for journal publications and future grant applications.'

Using the term 'developers' (abstract and background) was not clear enough to know who you were describing and studying. Also, it would be helpful to establish earlier in the paper that these interventions are complex health and healthcare delivery interventions (or however you might word this).

We have rewritten the text under 'objective' at the start of the Abstract to make it clear what was meant by 'developer'. It now reads 'Identify how individuals involved in developing complex health and health care interventions (developers), and wider stakeholders in the endeavour, such as funders, define successful intervention development and what factors influence how interventions are developed.'

In addition, in the Background section of the paper we have removed the term 'developer'. The first time we now use this term is at the start of the methods section. Here we clearly indicate who we are referring to when we use the term developer and wider stakeholder: 'The design was a qualitative interview study of individuals who had developed complex interventions to improve health (i.e. developers) and wider stakeholders. Wider stakeholders were individuals responsible for or affected by health-and healthcare-related decisions that may be informed by research evidence, for example, funding panel members, and patient and public involvement (PPI) members.'

Methods:

The methods should begin with a succinct review of the overall design. This might also be a good place to provide information on the larger study (I think this is INDEX) and your study, and how the 2 relate.

We have now inserted a subheading 'Overall design' at the start of the methods section and inserted the following text:

The design was a qualitative interview study of individuals who had developed complex interventions to improve health (i.e. developers) and wider stakeholders. Wider stakeholders were individuals responsible for or affected by health-and healthcare-related decisions that may be informed by research evidence, for example, funding panel members, and patient and public involvement (PPI) members. The study was pragmatic in nature and utilised the fact that employing a qualitative method would enable attention to be given to context and processes, and for participants to describe their views and experiences in detail.¹⁶

The overall purpose of the interviews was to gain detailed insight into the challenges of developing complex health and health care interventions. The INDEX study they were part of, consisted of a systematic methods overview of different approaches to intervention development,⁵ a review of journal articles reporting intervention development for specific studies, the qualitative interview study reported here, and a Delphi consensus exercise that the reviews and interviews fed into.

Throughout the INDEX study, the focus was on the development of complex interventions that could be delivered in public health, primary, secondary or social care settings. These interventions could include behaviour change interventions and interventions directed at health providers, patients and policy makers. It did not focus on medicines or invasive interventions, e.g. pills or devices.

Sampling: Perhaps a better description of the main study (INDEX) would help clarify who were the interviewees. I was not clear what was meant by 'wider stakeholders in the endeavour'. And whether these were people who had developed and tested, and published on interventions? – ie how did you know they developed an intervention?.

Having inserted the above text at the start of the method section, we have now defined what we meant by developer and wider stakeholder. In addition, we have now provide more details about who was interviewed under the heading The participants.

It would be helpful to clarify that these interventions to change health are in the context of health behaviour (smoking cessation) as well as interventions directed at health providers/patients/policy makers for the purpose of ensuring the care delivered was in line with best practices (following clinical practice guidelines). Also, what would define novice vs experienced? What is meant by intervention type and setting?

We have now inserted the following text at the start of the Methods section, under the subheading Overall design, to clarify what interventions we were interested in:

Throughout the INDEX study, the focus was on the development of complex interventions that could be delivered in public health, primary, secondary or social care settings. These interventions could include behaviour change interventions and interventions directed at health providers, patients and policy makers. It did not focus on medicines or invasive interventions, e.g. pills or devices.

We have also now indicated what we meant by intervention type and setting, deleted the text referring to novice and experienced, and have inserted the following text into the Recruitment and sampling section: 'We also aimed for diversity in terms of level of experience, sampling individuals who had completed just one intervention development, through to those who had contributed to multiple developments.'

Methods: the interviews were conducted by an experienced health services researcher – this is good but what specifically is meant by this? What kind of experience did this person have?

We have now inserted the following text: All interviews were conducted by NR, who has over 20 years' experience of conducting qualitative interview studies and evaluating health care technologies, but has limited experience of intervention development. This meant that she had a good understanding of the general context but probed on aspects of intervention development that might be taken for granted by a more experienced developer.

You indicated the guide was based on the aims of the study and relevant literature – it would be useful to have more details here – what exactly was the guide based on?

We have now inserted the following text: The guide was informed by the overall aim of the qualitative study and early findings from the INDEX systematic review of approaches to intervention development.⁵

Presumably you mean this study as opposed to the INDEX main study?

We have now clarified the guide was informed by the aims of the qualitative study (see response to prior comment). In addition, we have now indicated at the start of the Recruitment and sampling section that we are focusing on the qualitative interview study.

I thought the types of questions asked were extremely useful but I had a few questions: A topic guide was developed to 'ensure key areas were covered', but what were the key areas? One of the aims of the study (end of background) was to determine how they defined success so it seemed strange that this was not initially included in the guide – why not and what was initially included in the guide? Some sample questions here would be useful. Perhaps even relating the guide to the 2 specific aims for the study – ie how they defined success and the factors that influenced the process they chose.

We have now changed the text that it is clear what key areas the guide covered. It reads 'Key areas covered by the guide included participants' use of intervention development approaches, their experience of developing interventions, how they had developed and evaluated specific interventions, and what guidance and advice they would give to researchers undertaking this process.'

Initially we did not set out to specifically ask participants to define success as we were concerned that specifically asking participants to define successful intervention development would lead to them providing answers they thought would be socially desirable amongst the academic community. However, as we explain in the methods section of the paper, 'analysis of the first few interviews indicated definitions of success and drivers were important themes in the data.' We therefore added two new questions to the guide. We have changed the text so that these questions are now listed. The text reads 'Thus, after the first few interviews, the following questions were inserted into the guide: I'm interested in what counts as success in the context of intervention development. Do you think of this development study, as a successful study? Why?; Were there any key aims or principles guiding the development of the intervention? What and why?'

I was curious about the issue your interviewees had regarding anonymity. Was this determined after they agreed to participate or did this play a role in recruitment?

The information sheet given to individuals as part of the recruitment process stated that 'you will not be named or identifiable in any way' because anecdotal experiences of the team suggested this would encourage individuals to take part and to give more honest answers when interviewed. The need for anonymity was confirmed by a couple of the early participants - one person reviewed their transcript and asked for a small amount of text to be deleted, another person asked during the interview for specific comments to be considered "off the record". Following these two interviews, the team took the decision that only the interviewer would know the identity of participants, although the team did discuss potential interviewees and monitored characteristics of recruited participants closely to maximise diversity. When we explained this to later participants, we had no further requests for sections of interview to be redacted.

We have now explained some of this in the data collection section. At the start of this section, we have inserted the following text 'Potential participants were approached by email with an attached information sheet. The information sheet provided further details about the qualitative study, what participation would entail, and included a copy of the consent form that would be completed prior to interview. The sheet also informed individuals that if they took part, they would not be named or identifiable in any way. Providing anonymity was viewed as important because some participants would be well known within the research community.' At the end of this section we now state 'Early on during data collection, the importance of maintaining participant anonymity was confirmed; one person reviewed their transcript and asked for a small amount of text to be deleted, another person

asked during the interview for specific comments to be “off the record”. Following these interviews, the team took the decision that only NR would know who had been interviewed.’

Analysis: Again, more detail is needed. How did you test and refine the initial coding scheme? It seems this was done prior to applying it to an interview.

We have added much more detail to the analysis section. In addition, we have now explain that the analysis of the data presented in this paper occurred in two stages, and detailed more clearly how the coding framework was developed, refined and tested. The following text is now included in the Data analysis section:

‘The data were analysed thematically,¹⁸ so that comparisons could be made within and across the interviews, and participants’ views of specific issues highlighted, e.g. how individuals identified the need for a new intervention. To do this, a coding frame was drafted, tested and refined. NR (social scientist), PH (Academic General Practitioner), KT (social scientist) and ED (health service researcher and allied health professional) independently read three transcripts and constructed a coding frame. They then discussed their coding and combined their coding frames into a single framework. Each of them then applied this framework to two new transcripts from interviews conducted at different times in the data collection process. Another discussion was then held which resulted in the addition of further codes, and the deletion or clarification of some existing ones. The framework was then finalised, and applied by NR to transcripts using NVivo 11 software.¹⁹ Each member of the team applied the final coding frame independently to one transcript; the aim of this exercise was not to establish inter-rater reliability as all subsequent coding using this framework was conducted by NR but to allow comparison of, and reflection on, differences.

The framework applied by NR included the code ‘definitions of success’ and other codes that were relevant to the aims of this paper, e.g. development pathway, study endpoint, evaluating complex interventions. NR, KT and PH read data under these codes and discussed themes they had identified. As the framework NR applied had not been developed with the specific aims of this paper in mind, following these discussions KT re-read all the transcripts to check whether there were other relevant themes which needed to be captured. This also helped her contextualise and fully understand comments made. The re-reading of these transcripts led to KT developing a few more codes, e.g. indicators of success, actions needed, tensions. These codes were added to the framework used by NR and applied by KT where appropriate. Data coded under them were then discussed with NR and PH. Using an approach based on Framework analysis,²⁰ data relevant to this paper were summarised in a table formatted so that the rows represented each participant and columns relevant codes. KT then read and re-read the table’s content, reading down each column in order to identify similarities and differences between the views and experiences of different participants in relation to a specific code or theme, and across the columns to consider what might have influenced each participant’s views and experiences.’

What was meant by ‘how individuals identified areas for intervention’ - is this about the process of intervention design or about the health context that was being investigated, and if about the health context, how does this relate to the study aims of looking at the process of intervention design?

This text has been changed to: ‘how individuals identified the need for a new intervention.’

I am not sure how a thematic analysis allows you to compare across interviews. Typically a thematic analysis allows you to consolidate the main messages being found across all of the interviews.

We have added the following text to the data analysis section which should help readers understand how thematically analysis (and Framework analysis) allows you to compare across the interviews: 'KT then read and re-read the table's content, reading down each column in order to identify similarities and differences between the views and experiences of different participants in relation to a specific code or theme, and across the columns to consider what might have influenced each participant's views and experiences.'

I was not 100% certain what happened with the initial coding scheme once applied to the transcripts. Did you entertain changes to the scheme after the first few transcripts? I would like to see more about how the 'discussions at team meetings on the emerging analysis' shaped the final analysis.

We have now provide much more detail in the data analysis section which should address the first question. To address the second question, we have also inserted the following sentence 'At these meetings NR and KT detailed their insights and interpretation of the data, inviting other team members to comment in light of the aims of the analysis, and their knowledge of the data and literature on intervention development.'

Findings: I would like to see more detailed information on the participants. Who were the intervention developers vs the stakeholders? How were these each defined, and were there any differences in these in the results?. 'Most participants held various roles across different intervention developments' - this requires more detail. Did they develop an intervention alone, as part of a team, what specific experiences did they have. How many interventions? Did they test these interventions? Publish? How many of them follow specific development processes like intervention mapping or using the TDF and behaviour change wheel processes, MRC guidance? What is a 'target population centred' approach? Were any of them implementation scientists? There is not enough information on the participants.

We have now clearly defined developers and wider stakeholders at the start of the methods section, under the heading of Overall design.

There were not clear differences between the views of developers and wider stakeholders, apart from the fact that individuals interviewed as funding board members said there was flexibility in terms of the extent to which developers needed to define the potential intervention in their applications for grant funding, whereas some developers did not think this was the case. This is highlighted in the final paragraph of the section headed Securing funding to develop interventions.

To provide more detail about who was interviewed, we have expanded the text describing participants. It now reads:

Six of the 21 participants interviewed were wider stakeholders. Three of these stakeholders were funding panel members, two were PPI members, and the remaining stakeholder worked in health service implementation. In terms of the remaining participants, i.e. the developers, eight had led intervention development, four as 'senior leads' and four as early career researchers. Some of the senior leads had contributed to over 20 development projects. However, experience was not just about numbers of interventions completed but also about the size and complexity of the intervention. Senior leads had typically been responsible for developing highly complex interventions that would need considerable health service buy-in to implement, and had developed these interventions in the context of large scale, £1 million plus (or equivalent in other currencies) research funding. Early career researchers had typically completed smaller scale developments as part of a doctoral or post-

doctoral fellowship. Participants who had developed the most interventions typically had specialist skills, e.g. product design or behavioural science.

Except for one participant who had developed an intervention mainly on their own, participants had worked within a team, and most participants had held various roles across different intervention developments (e.g. lead on one, co-investigator on another). Only two participants (both on research funding panels) had no direct experience of developing interventions. Between them, participants had experience of a wide range of intervention development approaches, including theory based, partnership, target population-centred and combined approaches.⁵

The reference given at the end of this text explains what a target population-centred approach is, in addition to also explaining what theory based, partnership and combined approaches were.

Are the 'drivers of intervention development' the same as asking people about the factors that influence their processes (2nd study objective)? I think I would prefer the 'factors that influence how they develop their interventions'.

We appreciate the term drivers needed to be clarified. We now indicate early on in the paper that drivers are factors that shape and drive the development process., i.e. we state in the Data collection section that 'Initially the guide did not include specific questions about how successful intervention development should be defined, and what factors shaped and drove the development process, but analysis of the first few interviews indicated success and drivers were important themes in the data.' We have also edited the aim of the paper so it introduces the idea of driving factors: This paper aims to detail how those interviewed defined successful development and the factors they viewed as driving the development process.'

We are keen to keep the term driver as many psychological theories of motivation, goals and behaviour have built on the concept of drive (first cited by Maslow, A Theory of human motivation. Maslow AH. A theory of human motivation. Psychological review. 1943 Jul;50(4):370) and consider a range of overt or covert intrinsic and extrinsic motivations, rewards and threats to innovation or any proposed change. Factors is an inadequate term to encapsulate this and "drivers" is a commonly used term throughout the NHS when innovation or change is proposed. For example, it is a term used in the title and content of the following two 2018 BMJ group papers:

Brodersen J et al. Focusing on overdiagnosis as a driver to too much medicine. BMJ 2018; 362. doi: 10.1136/bmj.k3494.

Greszczuk C et al. Peer influence as a driver of technological innovation in the UK National Health Service: a qualitative study of clinicians' experiences and attitudes. BMJ Innovations 2018;4:68-74

The first theme is really interesting but it seems like a mix between the outcomes of intervention development (eg. must be cost effective), and the process itself (eg. consider sustainability early). This theme seems more about what a successful intervention is as about, as opposed to successful intervention development. I think some conceptual clarity on this would help.

We have addressed this comment, through changes we have made to this section, including inserting subheadings that reflect outcomes and processes of development, in response to the earlier comment that started 'An overall comment: It was unclear etc'.

I did not follow how ambition to improve health was a factor in their processes. Wouldn't most people who develop these interventions do so in order to improve health or health care delivery? This seemed more like the way in which people ended up being interested in health services research, not the factors that influence how they design these interventions once they decide to do so. It seems more about why they do it, as opposed to how.

We have changed the text at the start of this section to make it clearer that this ambition to improve health and health care delivery was often the starting point/a trigger to developing an intervention, and thus was a factor influencing the development process. It also drove what the purpose of the intervention would be. The text now reads 'When detailing where the initial idea for an intervention had come from, developers described how they had identified through their work a specific health or health care issue they wanted to improve. It was clear this ambition to improve a situation had been their motivation to develop the intervention, and that the issue they had identified had driven the intervention's focus.'

In addition, we have deleted some of the text in this section which did not appear relevant in terms of driving the development of a specific intervention.

The section on existing health contexts, care and legislation seems nicely aligned with the aims of the study and quite interesting to read, the other themes focused on the process of development were also very informative and useful (and aligned with my assumptions about what the study was about). The issue of relying on RCT's is again, an important factor but it is not connected to development per se (unless you were able to outline how this imperative for evaluation has an effect on how they designed the intervention).

Thank you for these positive comments. We connected the need for RCT evidence to intervention development by explaining that 'Several of these participants talked about there being a tension between developing interventions that were suitable for use in the real world and developing interventions that could be evaluated in an RCT', and that 'this drive for evidence could work against the goal of developing useful real-world interventions.' However, we appreciate this connection could be clearer, and therefore have inserted new text (underlined) into the first paragraph under the subheading so that it now reads 'Participants talked about public health and health care practitioners, and purchasers of health care services, wanting RCT-based evidence that demonstrated the intervention's effectiveness and cost-effectiveness before they were willing to support or purchase it. This meant developers felt a need to develop interventions that could be evaluated within a RCT. Several participants viewed this as problematic, explaining that they felt there was a tension between developing interventions that were suitable for use in the real world and developing interventions that could be evaluated in an RCT.'

Discussion:

The principal findings should be revised given points made above for the findings, and could be shortened, giving more room to look at other literature.

We have edited and shortened the section detailing the principal findings. It now reads: 'Developers and wider stakeholders defined successful intervention development as a rigorous, scientific process that resulted in effective interventions that could be implemented in real world contexts. The drivers that shaped how interventions were developed indicated tensions arose when trying to meet the needs of existing structures. Developers targeted areas where change could result in better health outcomes. Research evidence and theory informed the content, structure and delivery of interventions to increase the likelihood of them being effective. The characteristics of health contexts also shaped

the structure and content of the interventions developed, although here the drive was mainly to ensure they would be acceptable and feasible to deliver. Yet, as UK participants talked about how resource limited these health contexts could be, this driver also encouraged developers to be pragmatic and realistic in their designs, developing interventions that were not innovative or resource-intensive, possibly reducing their potential effectiveness. Perceived requirements of funding bodies also discouraged innovation, and the reliance on RCTs to prove effectiveness worked against the need for real-world relevance.'

Weaknesses of the study: I was not sure what was meant by the interview process allowing interviewees to say what they thought, as opposed to the research team. Also it would be wise to indicate that a limitation is only have one coder (if I am right about that).

We have now shortened this sentence, removing the reference to the research team as this is not needed. It now reads 'The in-depth and flexible nature of the interviews... allowed participants to describe, in detail, issues important to them.' Having expanded the text in the Methods section to detail more clearly how the coding frame was developed, it should now be clear that various individuals were involved in its development, refinement and testing.

Diagram 1: I was not fully able to interpret the diagram. How were short, medium and long term determined? Why would acceptability to stakeholders only be a short term goal? Is cost effectiveness only medium and not long?

We have inserted the follow text to support interpretation of the diagram: 'The definitions they [participants] gave suggested they thought about short, medium and long-term goals, all of which they might consider from the start of the development process but for which evidence to indicate they had been achieved would come at different time points in the future (diagram 1).'

In addition, we have changed the title of the diagram and the subheadings within it. The title is now 'Definitions of success identified in the data', and the subheadings are 'Short term definitions relating to processes and outcomes'; 'Medium term definitions relating to outcomes'; 'Long term definitions relating to outcomes.'

The reporting checklist indicates that grounded theory was used in the design, but this is not mentioned anywhere in the paper itself. Also, that 4 coders were used but this was only for one transcript, it seems that most of the coding was done by 1 person (but I could be wrong).

We did not use grounded theory. In the COREQ form we stated 'The study drew on the principles of grounded theory (our underline) in terms of insights from earlier data collection informing the focus of later interviews.' As this statement was also questioned by reviewer 2, it has now been changed to COREQ form so that it states 'This was a pragmatic study that was not underpinned by a specific methodological orientation or theory. It did, however, draw on some of the principles of grounded theory, in the sense that early data collection influenced some of the later interviews.'

Having provided more detail in the Data analysis section about the coding frame, it should now be clear that whilst only NR electronically coded transcripts in NVIVO, the coding frame she applied had been developed through discussions with 3 other individuals who had read and coded various transcripts manually. In addition, we have added a section explaining that for this paper, further

analysis was conducted specifically for this paper. This analysis was led by KT but supported through discussions with other team members.

Reviewer: 2

Reviewer Name: Dr Joy Probyn

Institution and Country: University of Salford, UK

Please state any competing interests or state 'None declared': None declared

Please leave your comments for the authors below

This is an interesting study that certainly makes a contribution to a gap in knowledge and will be a useful resource for researchers working in intervention development. There are some aspects of the paper which I feel merit improvement - all related to the methodology, outlined as follows:

Thank you for your positive comments and your helpful suggestions on how the paper could be improved.

There is no reference to qualitative epistemology/theory as justification for the methods chosen. You say on the COREQ checklist that it was a grounded theory study - this needs to be explained and linked to the explanation of methods.

We have inserted the following text under the heading Overall design: The study was pragmatic in nature and utilised the fact that employing a qualitative method would enable attention to be given to context and processes, and for participants to describe their views and experiences in detail.¹⁶

In the COREQ form we had stated 'The study drew on the principles of grounded theory (our emphasis added) in terms of insights from earlier data collection informing the focus of later interviews.' We do not state this was a grounded theory study but appreciate this sentence might have been misleading. We therefore have changed the COREQ form so that it states 'This was a pragmatic study that was not underpinned by a specific methodological orientation or theory. It did, however, draw on some of the principles of grounded theory, in the sense that early data collection influenced some of the later interviews'. Where we drew on the principles of ground theory is indicated in the data collection section of the paper where we state 'Initially the guide did not include specific questions about how successful intervention development should be defined and what factors shaped and drove the development process, but analysis of the first few interviews indicated definitions of success and drivers were important themes in the data. Thus, after the first few interviews, the following questions were inserted into the guide: I'm interested in what counts as success in the context of intervention development. Do you think of this development study, as a successful study? Why?; Were there any key aims or principles guiding the development of the intervention? What and why?'

There are no references at all in the methodology section.

We have now referenced purposeful sampling, thematic analysis, NVIVO 11, framework analysis, and data saturation, using the following references:

Patton MQ. Qualitative research and evaluation methods. Sage Publications, 2002.

Braun V, Clarke V. Using thematic analysis in psychology. Qual Res Psychol 2006;3:77-101.

NVivo qualitative data analysis Software; QSR International Pty Ltd.

Ritchie J, Spencer L. Qualitative data analysis for applied policy research. In: Bryman A, Burgess RG, editors. Analyzing Qualitative Data. London: Routledge; 1994. p.173-194.

Saunders B, Sim J, Kingstone T et al. , Baker S, Waterfield J, Bartlan B, Burroughs H, Jink C. Saturation in qualitative research: exploring its conceptualization and operationalization. Qual Quan 2018;52:1893-1907

There needs to be a better explanation of how the interviews were designed and how this links to data analysis. It appears that the data analysis was deductive - i.e. the two themes were predetermined and the data was 'fitted' in to these categories. This does not fit with a grounded theory approach.

In response to comments from the other reviewer, we have now detailed the overall aim of the interviews and where they occurred within the wider INDEX study. We have also explained that the topic guide was informed by the aim of the qualitative study and early findings from the INDEX methodological review (now described at the start of the methods section). In addition, we have expanded the data analysis section to indicate the inductive nature of the analysis undertaken for this paper.

As explained above, this study did not adopt a grounded theory approach but drew on its principles when undertaking data collection.

The findings read as quite obvious/descriptive for this reason - and theme 2 seems to be more inductively generated than theme 1, with the use of sub-headings. The two themes are actually very similar and I think it would improve the work to conduct the analysis again, inductively, and present the themes in this way. This would add rigour to the study. For example, some inductive themes could be 'sustainability', 'dependability' etc. A deductive analysis is fine - but it needs to be explained in the methodology more clearly if so.

Both themes were inductively generated and now include subheadings. We have expanded the section on data analysis to provide more detail of how the data were analysed inductively. The findings have been written in a way that indicates they are based on participants' accounts and the issues they raised.

There needs to be an explanation of how informed consent was obtained.

We have now included the following text at the start of the Data collection section: 'Potential participants were approached by email with an attached information sheet. The information sheet provided further details about the qualitative study, what participation would entail, and included a copy of the consent form that would be completed prior to interview.' Later on we state 'participants completed the consent form (verbally for telephone/skype and written for face-to-face).'

Was there any involvement from the public in the design and conduct of the research?

At the request of the editor we have now include in the methods a section on 'Patient and Public Involvement'. It reads 'Patient and Public Involvement (PPI) members were invited to an expert panel meeting held at the start of the INDEX study, and to a consensus conference held once the Delphi results were known. None attended the first meeting, but PPI members did attend the consensus conference aimed at interpreting the Delphi results. All study participants will be sent copies of journal publications resulting from the INDEX study.'

P7 44-45 suggests a grounded theory influence - explain.

We think this comment relates to the fact that we explain in the paper that later questions were added to the guide, following analysis of early interviews These questions have now been added to the text.

Where did the coding frame come from?

In the Data analysis section we have now provided much more detail about how the coding frame was developed through different members of the research team independently reading and coding transcripts, and then discussing their coding.

P 38 - Reference needed for data saturation.

We have now inserted the following reference: Saunderson B, Sim J, Kingstone T, Baker S, Waterfield J, Bartlan B, Burroughs H, Jink C. Saturation in qualitative research: exploring its conceptualization and operationalization. Qual Quan 2018;52:1893-1907.

P8 36-37 Why did people decline you say they were based outside of the UK but was this the reason they gave or your assumption? It shouldn't matter with the use of Skype?

Decline was implicit (they did not reply) rather than explicit. We have now made this clearer in the text by inserting the following 'Most of those who did not respond to the invitation were outside the UK.'

P8 54 What was the geographical spread of those invited?

We now state in the text, under Recruitment and sampling, that we sampled individuals from the UK, other European countries and North America. In addition, in table one (which provides information on those interviewed) we indicate participants were from UK, other parts of Europe, and North America.

VERSION 2 – REVIEW

REVIEWER	Heather Colquhoun University of Toronto, Canada.
REVIEW RETURNED	12-Apr-2019

GENERAL COMMENTS	The authors have responded to the reviews in a thoughtful and comprehensive manner. I have no further comments.
---